# Primary Cardiac Intimal Sarcoma: Multi-Layered Strategy and Core Role of *MDM2* Amplification/Co-Amplification and *MDM2* Immunostaining

**DOI:** 10.3390/diagnostics14090919

**Published:** 2024-04-28

**Authors:** Claudiu Nistor, Camelia Stanciu Gavan, Adelina Birceanu, Cezar Betianu, Mara Carsote, Anca-Pati Cucu, Mihaela Stanciu, Florina Ligia Popa, Adrian Ciuche, Mihai-Lucian Ciobica

**Affiliations:** 1Department 4-Cardio-Thoracic Pathology, Thoracic Surgery II Discipline, “Carol Davila” University of Medicine and Pharmacy, 050474 Bucharest, Romania; claudiu.nistor@umfcd.ro (C.N.); adrian.ciuche@umfcd.ro (A.C.); 2Thoracic Surgery Department, “Dr. Carol Davila” Central Military Emergency University Hospital, 010242 Bucharest, Romania; camigavan@gmail.com (C.S.G.); anca-pati.cucu@drd.umfcd.ro (A.-P.C.); 3Pathology and Immunohistochemistry Laboratory, Pathoteam Diagnostic, 051923 Bucharest, Romania; adelinabirceanu@gmail.com; 4Department of Interventional Imaging, “Doctor Carol Davila” Central Military Emergency University Hospital, 010825 Bucharest, Romania; dr.cezarbetianu@gmail.com; 5Department of Endocrinology, “Carol Davila” University of Medicine and Pharmacy, 050474 Bucharest, Romania; 6Department of Clinical Endocrinology V, C.I. Parhon National Institute of Endocrinology, 020021 Bucharest, Romania; 7PhD Doctoral School, “Carol Davila” University of Medicine and Pharmacy, 020021 Bucharest, Romania; 8Department of Endocrinology, Faculty of Medicine, “Lucian Blaga” University of Sibiu, 550024 Sibiu, Romania; mihaela.stanciu@yahoo.com; 9Department of Physical Medicine and Rehabilitation, Faculty of Medicine, “Lucian Blaga” University of Sibiu, 550024 Sibiu, Romania; florina.popa@yahoo.com; 10Department of Internal Medicine and Gastroenterology, “Carol Davila” University of Medicine and Pharmacy, 020021 Bucharest, Romania; lucian.ciobica@umfcd.ro; 11Department of Internal Medicine I and Rheumatology, “Dr. Carol Davila” Central Military University Emergency Hospital, 010825 Bucharest, Romania

**Keywords:** heart surgery, dyspnoea, cardiac sarcoma, ultrasound-guided cardiac coaxial needle biopsy, immunohistochemistry, *MDM2*, gene testing

## Abstract

Primary cardiac tumours are relatively uncommon (75% are benign). Across the other 25%, representing malignant neoplasia, sarcomas account for 75–95%, and primary cardiac intimal sarcoma (PCIS) is one of the rarest findings. We aimed to present a comprehensive review and practical considerations from a multidisciplinary perspective with regard to the most recent published data in the specific domain of PCIS. We covered the issues of awareness amid daily practice clinical presentation to ultra-qualified management in order to achieve an adequate diagnosis and prompt intervention, also emphasizing the core role of *MDM2* immunostaining and *MDM2* genetic analysis. An additional base for practical points was provided by a novel on-point clinical vignette with MDM2-positive status. According to our methods (PubMed database search of full-length, English publications from January 2021 to March 2023), we identified three studies and 23 single case reports represented by 22 adults (male-to-female ratio of 1.2; male population with an average age of 53.75 years, range: 35–81; woman mean age of 55.5 years, range: 34–70) and a 4-year-old child. The tumour-related clinical picture was recognized in a matter of one day to ten months on first admission. These non-specific data (with a very low index of suspicion) included heart failure at least NYHA class II, mitral regurgitation and pulmonary hypertension, acute myocardial infarction, ischemic stroke, obstructive shock, and paroxysmal atrial fibrillation. Awareness might come from other complaints such as (most common) dyspnoea, palpitation, chest pressure, cough, asthenia, sudden fatigue, weakness, malaise, anorexia, weight loss, headache, hyperhidrosis, night sweats, and epigastric pain. Two individuals were initially misdiagnosed as having endocarditis. A history of prior treated non-cardiac malignancy was registered in 3/23 subjects. Distant metastasis as the first step of detection (*n* = 2/23; specifically, brain and intestinal) or during follow-up (*n* = 6/23; namely, intestinal, brain and bone, in two cases for each, and adrenal) required additional imagery tools (26% of the patients had distant metastasis). Transoesophageal echocardiography, computed tomography (CT), magnetic resonance imagery, and even ^18^F-FDG positronic emission tomography-CT (which shows hypermetabolic lesions in PCIS) represent the basis of multimodal tools of investigation. Tumour size varied from 3 cm to ≥9 cm (average largest diameter of 5.5 cm). The most frequent sites were the left atrium followed by the right ventricle and the right atrium. Post-operatory histological confirmation was provided in 20/23 cases and, upon tumour biopsy, in 3/23 of them. The post-surgery maximum free-disease interval was 8 years, the fatal outcome was at the earliest two weeks since initial admission. *MDM2* analysis was provided in 7/23 subjects in terms of MDM2-positive status (two out of three subjects) at immunohistochemistry and *MDM2* amplification (four out of five subjects) at genetic analysis. Additionally, another three studies addressed PCISs, and two of them offered specific *MDM2*/*MDM2* assays (*n* = 35 patients with PCISs); among the provided data, we mention that one cohort (*n* = 20) identified a rate of 55% with regard to *MDM2* amplification in intimal sarcomas, and this correlated with a myxoid pattern; another cohort (*n* = 15) showed that MDM2-positive had a better prognostic than MDM2-negative immunostaining. To summarize, *MDM2* amplification and co-amplification, for example, with *MDM4*, CDK4, *HMGA3*, *CCND3*, *PDGFRA*, *TERT*, *KIT*, *CCND3*, and *HDAC9*, might improve the diagnosis of PCIS in addition to *MDM2* immunostaining since 10–20% of these tumours are MDM2-negative. Further studies are necessary to highlight *MDM2* applicability as a prognostic factor and as an element to be taken into account amid multi-layered management in an otherwise very aggressive malignancy.

## 1. Introduction

Primary cardiac tumours are relatively uncommon, with benign myxomas accounting for approximately 75% of all cases [1]. Among the other 25% of the tumours, representing malignant masses, cardiac sarcomas account for 75–95%, with a severe prognosis, having a 5-year survival rate of 14% [1,2]. Generally, sarcomas represent a heterogeneous group underlying a variety of aetiologies and anatomical features [2]. According to the World Health Organization (WHO), angiosarcoma is the most common type of sarcoma (40%) followed by rhabdomyosarcoma (5%) and, less frequently, liposarcoma, osteosarcoma, leiomyosarcoma, as well as fibrosarcoma [3,4]. Intimal sarcomas of primary cardiac origin are extremely rare mesenchymal tumours across these mentioned histological types [1,2]; they develop in the pulmonary trunk, vein, and artery, as well as the aorta, coronary artery, and cave vein [5,6]. Left heart sarcomas usually infiltrate the posterior myocardium, resulting in valvular dysfunction, whereas right heart neoplasia is typically exophytic [5].

The management of primary cardiac tumours is complex, starting with a fine index of suspicion and followed by a multimodal imagery panel in association with tools of direct tumour access, which are mandatory for a better outcome, especially in individuals that are surgery candidates with intention of complete tumour removal [1,2,3,4]. After admission to secondary or tertiary highly specialized cardio-thoracic centres, a multitude of investigations are required in order to provide a specific diagnosis. Non-interventional methods such as transoesophageal/transthoracic echocardiography, computed tomography (CT), magnetic resonance imagery (MRI), and positronic emission tomography-CT (PET-CT) are essential in addition to interventional procedures (for example, cardiac catheterization with biopsy under CT guidance or 3D echocardiography-guided transthoracic coaxial needle biopsy). Overall, a multi-layered approach to this neoplasia allows for image-guided biological tissue sampling, which represents a major step in the management [1].

Currently, there are no specific treatment guidelines for primary cardiac intimal sarcomas because of the rarity of these cases and their heterogeneous spectrum [5]. Surgical removal of the tumour (potentially associated with auto-transplantation or heart transplant) is still the best treatment with radical intent [6] in small tumours without loco-regional/distant spreading or severe impairment of cardiac function [7,8,9,10,11]. Nonetheless, the lack of particular symptoms may delay the diagnosis as the patient might remain for a prolonged period of time only under primary health care surveillance and thus, the opportunity window for a successful intervention might be missed [5,8,9,10,11]. Additionally, chemotherapy, radiotherapy, immunotherapy, or molecular treatments with genomic agents are used as adjuvant/neo-adjuvant lines, as an alternative to surgery, in addition to complete/incomplete tumour removal or in recurrent post-operatory cases [5,12,13,14,15,16,17,18].

Deeper modern insights into approaching the topic of intimal sarcoma involve the *MDM2* (mouse double minute 2 homolog) profile in terms of *MDM2* gene amplification and co-amplification of other genes and positive *MDM2* immunostaining amid tumour tissue analysis that helps the identification of this histological subgroup among other sarcomas and the decision of further management [12,13,14,15,16,17,18].

We aimed to present a comprehensive review and practical considerations from a multidisciplinary perspective with regard to the most recent published data in the specific domain of primary cardiac intimal sarcoma. We covered the issues of awareness amid daily practice clinical presentation to ultra-qualified management in order to achieve an adequate diagnosis and prompt intervention, also emphasizing the core role of *MDM2* genetic analysis and *MDM2* immunostaining. An additional base for practical points was provided by a novel on-point clinical vignette.

## 2. Methods

We searched the PubMed database for the most recent full-length, English publications from January 2021 to 20 March 2024 and only included original articles in terms of studies and case reports with respect to the diagnosis of “primary cardiac intimal sarcoma”. No other histological types of cardiac tumours were analysed. We identified 81 papers from inception to the present time and searched the most recent 30 papers according to the mentioned time frame of publication. Finally, after manually checking each of these articles, 26 papers were selected (one article was excluded because it was not written in the English language, another was a review, and the other two introduced other histological types than intimal sarcoma) (Figure 1).

The new case report offered a practical multidisciplinary rationale for further discussing this challenging topic. In this instance, the patient parameters were retrospectively collected after receiving the approval of the Local Ethical Committee, as shown at the end of this paper.

## 3. Review of Published Data

According to our methods, we identified a total of 26 papers [19,20,21,22,23,24,25,26,27,28,29,30,31,32,33,34,35,36,37,38,39,40,41,42,43,44] (Table 1).

### 3.1. Epidemiological Input: Awareness of an Exceptional Finding

Generally, this histological type remains at the case report level of statistical evidence. We identified 23 novel single case reports (we did not include other presentations than intimal sarcomas) [19,20,21,22,23,24,26,27,29,30,31,32,33,34,35,37,38,39,40,41,42,43,44] and three studies [25,28,36]. The first group introduced 22 adults (male-to-female ratio of 1.2; male population with an average age of 53.75 years, range: 35 to 81 years; woman mean age of 55.5 years, range: 34 to 70 years) and a 4-year-old child [19,20,21,22,23,24,26,27,29,30,31,32,33,34,35,37,38,39,40,41,42,43,44].

The mentioned studies followed different methods of tumour exploration [25,28,36], and two of them [25,36] specifically provided data for intimal sarcomas (*n* = 35). One cohort [25] retrospectively analysed 48 cases of primary cardiac sarcomas amid *MDM2* testing, and fewer than half were intimal sarcomas (*n* = 15/48 patients; mean age of 47.2 years; range: 12–78 years) [25]. A second study explored the SEER database (The Surveillance, Epidemiology, and End Results) with respect to cardiac sarcoma presentation and outcome (no specific data were provided in terms of the age or sex ratio with respect to the intimal type) [28], while the third cohort included a re-do *MDM2* exploration in terms of gene amplification, histological profile, and immunostaining (*n* = 20 patients with primary intimal sarcomas) [36].

As mentioned, the youngest patients were 12 [25] and 4 years old [24]—the 4-year-old being a subject admitted for severe right heart failure that required cardiac surgery soon after initial admission. Complete tumour resection of the primary intimal sarcoma allowed for a good prognosis since the patient was followed for another 8 years and showed no recurrence while post-operatory adjuvant chemotherapy was declined by the parents [24].

### 3.2. Clinical Presentation

The tumour-related clinical picture was recognized in a matter of one day to a few days [39,44], two months [20], three months [30,34,35], six (or “several”) months [21,27], and ten months [40]. Data on admission included heart failure [19,24,42,43,44], either acute or recently detected [at least NYHA (New York Heart Association) class II], mitral regurgitation and pulmonary hypertension [21], acute myocardial infarction [29], ischemic stroke [31], obstructive shock [39,41], and paroxysmal atrial fibrillation [37]. Additionally, awareness might have come from other non-specific complaints, as pointed out by the original authors, such as (most common) dyspnoea [21,23,27,30,32,39,41], palpitation [21,35,40], chest pressure [23,35], cough [30,40], asthenia [33], sudden fatigue [39], weakness [43], malaise [32], anorexia [33,44], weight loss [33,40], headache [39], hyperhidrosis [40], night sweats [32], facial and peripheral oedema [20], and epigastric pain [44].

Two individuals were initially misdiagnosed as having endocarditis [22,42]. A history of malignancy was registered in some patients (*n* = 3/23) as follows: breast [23], melanoma [23,27], and cervical cancer [30]. Distant metastasis causing the detection of a cardiac tumour only as a secondary step amid whole body investigations was involved in two out of the twenty-three cases, specifically, brain [26] and intestinal [38] spreading. During follow-up, one of these two subjects again showed digestive metastases following primary tumour and first metastasis removal [38], another two individuals experienced bone metastasis [30,37], and another one was found with brain metastasis [43], with unilateral adrenal spreading with post-operatory histologic confirmation [44]. Hence, a total of six patients (*n* = 6/23, representing 26% of the cohort on published cases) were confirmed with distant metastatic cardiac sarcoma (outside the cardio-thoracic and mediastinal area) [26,30,37,38,43,44].

### 3.3. Imagery Tools of Diagnosis

Imaging procedures are essential for sarcoma exploration, as is similarly seen in other cardio-thoracic malignancies [45,46,47]. They include traditional tools such as cardiac ultrasound like (most importantly) transoesophageal, as well as a transthoracic approach, CT, and MRI. The misdiagnosis as myxoma or prior diagnosis of unrelated cancers required investigation of distant areas [19,20,21,22,23,24,25,26,27,28,29,30,31,32,33,34,35,36,37,38,39,40,41,42,43,44].

According to the sample-focused analysis (*n* = 23), the tumour size varied from 3 cm to masses larger than 9 cm. For instance, we mention the maximum diameters (in cm) of 3 [23,27], 3.7 [40], 3.9 [41], 4 [44,48,49], 4.2 [19], 4.7 [37], 5.8 [37], 6.7 [30] 7 [26], ≥9 [20,21,34], and, overall, a mean largest diameter of 5.5 cm (but a potential bias in size collection might come from different imagery procedures or data collection). The most frequent site was left atrial [21,30,31,32,37,39,40,41,44] followed by right ventricular [20,23,24] and right atrial [33]. ^18^F-fluorodeoxyglucose (^18^F-FDG) PET-CT offered a fine index of information since hypermetabolic lesions are registered in primary cardiac sarcomas, as opposite to benign lesions (which are more frequent); however, the distinction with secondary malignant involvement is not clear [21,24,29,32,35]. For instance, one reported case of a senior female showed the importance of this differential diagnosis amid ^18^F-FDG PET-CT use; yet, this came as an additional pitfall since the subject was already diagnosed with a previous non-cardiac malignancy [23]. In addition to this aspect, the presence of intra-tumour calcifications and ossification-like patterns might mimic mixomas, and they should also be differentiated from long-standing intra-cardiac thrombi [23]. Similarly, in 2023, Bergonzoni et al. [22] showed another conundrum of imagery aspects on a 73-year-old male with a mitral valve sarcoma who was initially suspected of having endocarditis based on clinical and ultrasound findings [22]. In short, PET-CT might indicate a hypermetabolic status (prone to a malignancy) at first admission [21,35] or during follow-up (highlighting metastasis) [44].

### 3.4. Management and Outcome

A longer free survival rate was achievable in a minority of the cases, and this comes from early detection and prompt surgery under cardiopulmonary bypass in an otherwise very aggressive malignancy [21]. A median survival rate of 17–24 months is generally reported [26,27]. Surgical removal of the neoplasia stands for the gold standard of therapy [27]. Early complications following en bloc or debulking resection are expected including coagulation anomalies, severe inflammation, thrombosis, recurrent malignant arrhythmias, massive bleeding, acute disseminated intravascular coagulation, and tumour lysis syndrome with fulminant outcome [20]. Pathogenic traits of these early post-surgery complications are linked to tumour features and surgical and anaesthesia processes, as well as cardiovascular and hemodynamic changes across neoplasia removal [20,21]. They can be hardly predicted before surgery; however, clues such as a mildly reduced number of pre-operatory blood thrombocytes or severe cardiac dysfunction should be taken into consideration [20]. Meticulous early follow-up after tumour resection also includes coagulation and inflammatory profile surveillance, but, as mentioned, there is no standard protocol, only an individualized decision. Taking into account the medical background, inflammatory, vascular, autoimmune co-morbidities, and prior medication in one patient is mandatory since they might influence the overall outcome [48,49,50,51].

Cases that are not surgery candidates or display post-operatory recurrence may be submitted to anthracycline protocols of chemotherapy. Alternatively, pazopanib (tyrosine kinase inhibitor) has been used. Chronic renal failure, for instance, due to tumour-related burden or prior diagnosis of cardiac insufficiency requires a switch to a pazopanib as first-line palliative therapy [19,22].

In the study by Cho et al. [25] (*n* = 48), multimodal management included the following: complete tumour resection (*n* = 11/48), partial excision (*n* = 23/48), cardiac biopsy or pericardiotomy (*n* = 10/48), cardiac transplant (*n* = 4/48), neoadjuvant chemotherapy or radiotherapy (*n* = 4/48), and adjuvant chemotherapy or radiotherapy (*n* = 34/48). Follow-up data (available for 34/48 patients) were available for an average of 15.9 months, and 85.3% of the patients died after a mean of 19.8 months. The panel of prognostic markers involved the following: MDM2-positive status (*n* = 15/48 with primary cardiac intimal sarcomas, including seven subjects who were newly reclassified amid re-do *MDM2* analysis) was correlated with left atrium site (*p* = 0.001). Complete versus incomplete resection offered a better prognosis (*p* = 0.045), yet cardiac transplant was not associated with a better outcome (*p* < 0.5). The survival rate in angiosarcomas was similar when compared to intimal sarcomas; *MDM2* positivity was better than MDM2-negative status (*p* = 0.003) in terms of outcome. Survival was not correlated with the tumour location or the application of neoadjuvant treatment. Adjuvant therapy improved the prognosis in angiosarcomas (*p* < 0.001) but not in intimal sarcomas (*p* > 0.5) [25].

Khan et al.’s [28] study focused on cardiac sarcomas (of all types) according to the SEER database; the TCGA (The Cancer Genome Atlas) database was used as a comparison (U.S. population). The cohort (*n* = 400, average age of 47.7 years) revealed that most cardiac sarcomas were undifferentiated (61.7%); 71% of them did not associate distant metastasis, 29% of the sarcomas were poorly differentiated, while many of this sub-group were described as intimal sarcomas originating from the pulmonary trunk. Surgery represented the most common first line of intervention and provided a better outcome (hazard ratio of 0.391, *p* = 0.001) than individuals only receiving chemotherapy (hazard ratio of 0.423, *p* = 0.001). Generally, patients younger than 50 years have a better survival rate than older subjects [28].

Our sample-focused analysis (*n* = 23) showed that post-operatory histological confirmation of the primary intimal sarcoma was performed in twenty cases [19,20,21,22,24,26,27,30,31,32,37,38,39,40,41,42,43,44] and upon tumour biopsy in three individuals [22,24,35]. Complete heart tumour resection was identified in 13 patients (regardless of post-surgery outcome) [24,27,28,30,31,32,37,39,40,41,42,43,44], including secondary non-cardiac surgery for brain [26] and adrenal [44] metastasis resection. Other first-line decisions included debulking surgery (*n* =2/23) [20,38] and no surgery, but radiotherapy (the patient refused chemotherapy) [23], post-biopsy chemotherapy [34], or immune therapy (5 out of 13 planned cycles with PD-1 antibodies) [35].

Post-operatory local recurrence was described as follows: after 3 months, also followed by a second surgery, then chemotherapy, and then adrenalectomy after one year [44]; after 4 months and when radiotherapy was started with further distant metastases registered after 18 months (no tolerance to chemotherapy) [38]; and after 6 months and a second cardiac surgery was performed, then the patient refused radiotherapy (and died after 6 months) [41]. Additionally, the post-surgery free disease interval was confirmed (*n* = 3) for 20 months [21], 8 years [24], and also for one year followed by the identification of bone metastases requiring adjuvant therapy [37].

The post-operatory chemotherapy panel included a heterogeneous spectrum of approaches, for instance, pazopanib (13 months) [19] or anthracycline-based drugs [22], planned ifosfamide with radiotherapy [26], chemotherapy for one year followed by radiotherapy because of local recurrence after three years [29], doxorubicin plus isophosphamide followed by local recurrence after 4 months [26], or chemotherapy for newly detected bone metastases [30,37]. Fatal outcomes were specifically registered following surgery at 3 months [31], 11 months [22], and 18 months [19] after 8 months after the second cardiac surgery [41]. The most aggressive disease evolution was found in two cases as follows: one person died “soon” after tumour resection [43] and another within two weeks after sarcoma biopsy [34].

## 4. Discussion

### 4.1. A Novel Case Identification

This was a 65-year-old female, without known cardiovascular risk factors, who was referred for rapidly progressive dyspnoea, while she was under the surveillance of her primary healthcare physician. Several imaging evaluations were then performed, which identified a left pulmonary tumour and suspected lung metastases. She had a history of surgery for bilateral breast cancer in addition to chemotherapy and radiotherapy that were performed 20 years before the current admission. Additionally, she presented thoracic pain, tachycardia, and cardiorespiratory failure. The electrocardiogram (ECG) revealed a normal sinus rhythm with a resting heart rate of 69 bpm, poor R wave progression, right bundle branch block ST elevation in leads V2–V6, I, a VL, and reciprocal ST depression in leads DIII and aVR, biphasic T waves in leads V2–V3 (Figure 2).

The troponin I level was found to be slightly elevated (90 ng/mL). Coronarography was used to rule out any luminal or anatomical abnormalities. A tumour filling the right ventricle and involving the apex of the left ventricle was identified at transthoracic echocardiography (Figure 3 and Figure 4).

The tumour size and degree of infiltration over the whole apex of the left ventricle were determined by injecting 2 mL of SonoVue (Bracco S.p.A., Milan, Italy) at the level of intra-myocardial contrast in the left ventricle. The presence of microbubbles at this level confirmed the vascularization of the tumour mass. A non-homogeneous right ventricular mass of 88/61/80 mm extending to the left ventricle with areas of necrosis in both ventricles, disseminated lobular nodules on both pulmonary fields measuring 9 mm (right lung), 17 mm (left lung), and right para-tracheal adenopathy were identified on a thoracic CT scan (Figure 5 and Figure 6).

An abdominal CT scan showed an enlarged liver with irregular margins and a multinodular structure. The cardiac tumour was additionally examined using MRI: the tumour mass was 7.5 cm transversely, 8 cm anteriorly, and 7.5 cm craniocaudally and exhibited diffuse inhomogeneous intensity. The right ventricular component was a mass of 48/22 mm with irregular borders and invasion of the interventricular septum, while the anterolateral wall component measured 25/27 mm and had an apical position and an irregular shape (Figure 7).

Since surgery was contraindicated by the general status of the patient, a 3D ultrasound-guided biopsy of the tumour was performed. The anatomical landmarks for the cardiac biopsy were established by integrating CT and MRI data. Low contractility of the cardiac apex allowed for the biopsy to be conducted at this level (Figure 8).

A complex multidisciplinary team supervised the biopsy that was performed in the operating room. Transthoracic ultrasonography using a convex probe identified the apical tumour, the absence of hypo-echogenic left ventricle motility in contrast with the remaining myocardial wall, and the infiltration of the adjacent pericardium. The biopsy was performed via the left inferior parasternal route by using an ultrasound-guided 10 cm 20G coaxial biopsy needle inserted percutaneously. Three tumour fragments of 1 by 2 cm were obtained. No incidents were recorded during or after the procedure. Histological reports confirmed the presence of a high-grade primary intimal cardiac sarcoma with areas of myxoid differentiation and epithelioid-appearing cells (Figure 9).

Immunohistochemistry analysis revealed a positive reaction for *MDM2* and a Ki67 proliferation marker of 20% to 80% in tumour cells (Figure 10). The samples were found negative with regard to myogenin, CD31, CD34, D2-40, and desmin. Despite the prompt individualized diagnosis, the patient died three weeks later.

### 4.2. Practical Points: From Awareness of Cardiac Tumours to Complex Management in Primary Intimal Sarcomas

This present case adds to the current limited publications with regard to primary cardiac intimal sarcoma, as mentioned [19,20,21,22,23,24,25,26,27,28,29,30,31,32,33,34,35,36,37,38,39,40,41,42,43,44], amid the general chapter of primary intra-cardiac malignancies [52,53,54]. Among primary heart tumours (affecting less than 0.1% of the general population according to autopsy series), an intimal sarcoma represents an exceptional finding [9]. It may be identified during investigations for other conditions, as reported by Chao et al. [25], underlying the traditional scenario of incidentalomas, as seen in other organs [55,56,57]. The tumour has a non-specific symptomatology, and it requires a high degree of suspicion to be diagnosed. Dyspnoea, as mentioned in the present vignette and sample-focused analysis [21,23,27,30,32,39,41], should be investigated as it represents one of the most useful clinical traits (yet, it remains highly non-specific) [25].

In 5% of the cases, cardiac sarcomas develop secondary to radiation for another type of cancer, especially for breast cancer [25], and, according to the literature, they are usually more aggressive tumours [58,59,60]. The latency period between the first tumour treated with radiotherapy and the occurrence of the cardiac sarcoma is a minimum of ten years (the average radiation dose associated with the development of sarcoma is approximately 50 Gy) [11]. Radiations induce genomic instability, resulting in the accumulation of harmful mutations and a subsequently increased risk of malignant transformation [11]. As seen here, a history of mammary cancer was identified by Awoyemi et al. [23]; other reports showed a prior diagnosis of melanoma [23,27] and cervical cancer [30]. Of note, the same patient had both malignancies (mammary cancer and melanoma) across our research [23]. Whether this represents a particular genomic configuration in a subgroup of oncologic patients amid the recognition of primary intimal sarcoma in the heart is an open issue. For example, Cowden syndrome underlying *PTEN* (phosphatase and tensin homolog) pathogenic variants has been reported to address both of the mentioned non-cardiac neoplasia (asynchronously or simultaneously in the same patient); yet, currently, no specific genetic syndrome has been described in cardiac intimal sarcomas [61,62,63]. Post-radiation sarcoma can emerge in any location in one patient, notably, in females who underwent radiotherapy following a mastectomy [12]. Sarcoma may be found close to the area targeted by radiotherapy [11,12]. In our patient, a cardiac sarcoma occurred 20 years after she underwent a bilateral mastectomy and two cycles of 50 Gy radiation therapies for each breast, and it should be listed as a potential contributor, yet with indeterminate significance because of the current level of statistical evidence in the particular matter of primary intimal sarcomas at heart. Moreover, cytogenetic abnormalities leading to the formation of cardiac sarcomas also include *K-ras* gene pathogenic variants [11,13]. Sarcomas exhibit two distinct forms of genetic changes as follows: complex karyotypes with rearrangements and non-specific chromosomal gains and losses and simple karyotypes with chromosomal-specific translocator events [11,64,65,66].

In this novel report, the presence of a cauliflower-shaped ill-defined cardiac tumour protruding inside the right ventricle, with an irregular shape, filling one-third of the anterior heart, situated apically, and infiltrating all the layers of the right ventricle’s anterior wall, was suggestive of a malignancy at MRI scan. The tumour infiltrated the septal, anterior, lateral, and apical walls of the left ventricle, resulting in myocardial necrosis. No cardiac thrombi were identified, and the mass did not expand to the valvular region. Notably, thrombus represents a very important differential diagnosis at first imagery evaluation, and it requires additional assessments [32]. Multimodal imaging for diagnosis (PET-CT, cardiac CT, and cardiac MRI) is beneficial in determining the benign versus malignant nature of primary cardiac tumours based on the tumour’s penetration into the tissue planes. Additionally, it can detect pleurisy, pericarditis, adenopathy, or metastases in relation to other pathologies [67,68,69].

In this vignette, the scenario of having cardiac metastasis from previous mammary malignancy could not be actually ruled out until histological confirmation. Imaging-guided percutaneous coaxial needle biopsies are extremely accurate at identifying malignant tumours in oncologic patients that associate tumours of various locations, and their results are enhanced by the sampling method’s accuracy. The indications for the improvement in the treatment strategies are correctly defined by oncologic boards, which include the oncologist, the surgeon, the cardiologist, the radiologist, the radiotherapist, the pathologist, etc. [14]. Because of the technique-dependent pathology diagnosis and immunohistochemistry profile, transthoracic coaxial needle biopsy represents a real challenge for the practitioner [14,70,71,72]. One of the particularities of this case on point was represented by performing an ultrasound-guided transthoracic coaxial needle biopsy of the heart. A tumour biopsy was performed to obtain the biological tissue needed for a specific diagnosis. However, if the tumour is located in the left heart, the procedure may become more challenging. To summarize, the current literature mainly describes two types of biopsy techniques for cardiac sarcoma including the following: trans-septal cardiac biopsy guided by 3D transoesophageal echocardiography [15,73,74,75] and intra-cardiac echocardiographic-guided cardiac biopsy [16,76,77,78], while in the above-mentioned analysis, we identified that 8.69% of the patients had a cardiac tumour biopsy as milestone of multi-layered management [22,24,35]. Notably, we identified a similar case published in 2022 by Salazar et al. [34] on a 65-year-old female who showed a 3-month history of dyspnoea and asthenia with weight loss. She was confirmed with a large cardiac tumour and underwent a transvenous biopsy with histological confirmation. Because of the severity of the clinical presentation, she was not considered a surgery candidate (as seen in the present case) and palliative chemotherapy was planned, but she died within two weeks after first admission [34].

Because of the rarity of primary cardiac intimal sarcomas and consequent lack of randomized controlled trials, there are currently no specific treatment guidelines. As already mentioned, surgical treatment, chemotherapy, radiation, molecular medicines targeting genetic alterations, immunotherapy targeting cell-mediated immune malignancy, transplantation, and auto-transplantation are all included in the therapeutic approach, but radical surgery can be performed only in selected cases. Likewise, we should mention that left heart sarcomas are typically located along the posterior wall of the left atrium (associated with exertional dyspnoea and congestive heart failure caused by obstruction of intra-cardiac blood flow), whereas right heart sarcomas are larger and exophytic in nature (being manifested with congestive heart failure induced by obstruction of intra-cardiac blood). Hence, medical intervention in addition to radical surgery in terms of controlling the cardiovascular, hemodynamic, coagulation, and inflammatory status is also very important [19,20,21,22,23,24,25,26,27,28,29,30,31,32,33,34,35,36,37,38,39,40,41,42,43,44].

### 4.3. Core Role of MDM2 Assays as First-Line Analysis and Re-Do Perspective

In this case, histological examination revealed an intimal sarcoma with areas of myxoid differentiation and epithelioid-appearing cells. The *MDM2* (immunohistochemistry-based) tumour marker is considered to be useful for high-grade intimal sarcomas nowadays. *MDM2* (an oncogene that is responsible for blocking the activity of p53) is defined by nuclear overexpression and amplification of the 12q12–15 area, which contains *CDK4* (cyclin-dependent kinase inhibitor 2A) and *MDM2* [17,18], the *MDM2* amplification supporting the diagnosis of intimal sarcoma [17]. Additionally, the intimal sarcoma with a positive stain for *MDM2* has been shown by some studies to be associated with a poor prognosis, with a median survival of three to twelve months [17,18]. In contrast, Cho et al. [25] showed that MDM2-positive show a better outcome than MDM2-negative tumours [25]. One more clue of the expected dramatic outcome is the increased value of the Ki67 proliferation marker at immunohistochemistry analysis, as seen in the present case (20% to 80% depending on tumour area), which is similar to other reports of different sarcomas such as osteosarcomas or leiomyosarcomas [17,18,79,80].

The discovery of *MDM2* over-expression in intimal sarcomas (as opposed to other histological subtypes) highlights the tumour signature at the molecular level, as it was first shown only a decade ago [81]. This breakthrough discovery led to a second wave of assays amid initial diagnosis based on tissue database reviews, which seems like the current approach in the matter of cardiac sarcoma (larger) studies, as previously mentioned according to our research on recently published data [25,36].

Re-do analyses in terms of *MDM2* amplification and co-amplification, for example, with *MDM4*, CDK4 (12q13–q15), *HMGA3* (12q13–q15), *PDGFRA* (4p12), *TERT* (telomerase reverse transcriptase) (5p15), *KIT*, *CCND3* (cyclin D3; on 6p21), and *HDAC9* (histone deacetylase 9), improves the diagnosis of primary cardiac intimal sarcomas in addition to *MDM2* immunostaining. Briefly, re-do *MDM2* analysis as a single input or in combination subscribes to the general oncologic frame focusing on molecular, genetic, and immunohistochemistry profiles. This is meant to provide new prognostic markers and clues for multimodal therapy response and hence, a better choice of anti-cancer treatment [36,82,83,84]. For example, Yamada et al. [36] re-analysed 20 cases of cardiac intimal sarcoma under the umbrella of re-do histological reports, immunohistochemistry for *MDM2*, ERG, alpha-smooth muscle actin (ASMA), desmin, respectively, AE1/AE3, and genetic testing (FISH) for *MDM2* and *PDGFRA* (platelet-derived growth factor receptor alpha). The histological profile showed the following: necrosis (75%); fibrinous deposits (70%); myxoid stroma and haemorrhage (each accounting for 60% of the case series); abnormal mitosis (50%), and rarely, a cartilaginous pattern (5%). Immunohistochemistry analysis showed focal positive assays for *MDM2* in 80% of cases, while other less frequent positive reactions were confirmed such as ASMA (30%) and desmin (25%), as well as AE1/AE2 and ERG (20% for each). Genetic testing showed *MDM2* amplification (55% of cases, *n* = 11/20), which correlated with the myxoid pattern (*p* = 0.0194) and *PDGFRA* amplification (5%). The site of the tumours did not correlate with any of the mentioned parameters. To conclude, the myxoid histological report was correlated with *MDM2* amplification and thus, intimal sarcoma sub-groups (myxoid and non-myxoid) might represent a new perspective of understanding different clinical behaviours since genetic and pathological backgrounds might be distinct [36].

In addition, one single-centric study based on 25 years of data (between January 1993 and June 2018) analysed heart sarcomas and provided a supplementary immunohistochemistry report for the *MDM2* profile as the main feature for the intimal type. Of 705 individuals who underwent cardiac surgery for various tumours, 6.8% (*n* = 48 patients) were confirmed with primary cardiac sarcomas (male-to-female ratio of 21 to 27). Their average age at diagnosis was 47.2 years, from 12 to 78. The panel of clinical elements included the following: dyspnoea in 45.8% of the patients (*n* = 22/48) of at least NYHA class II (as also mentioned in our case-based analysis [2,21,23,30,32,39,41]); chest pain in 16.7% of the subjects (*n* = 8/48) followed by cough (12.5%) and pericardial effusion (8.3%), as well as syncope (6.2%). An incidental scenario of detection was found in 4.2% of the entire cohort (*n* = 2). The site involved the right atrium in 52% of the subjects and the left atrium (31.2%). Across the histological types, intimal sarcoma was confirmed in 27.1% of the patients, and 84.6% of them were located within the left atrium (which came on second after angiosarcoma type that was confirmed in 47.9% of the entire cohort). By performing *MDM2* reaction amid immunohistochemistry exams, seven cases were re-classified to actually be intimal sarcomas (positive *MDM2* status) [25].

Genomic analysis in terms of *MDM2* amplification/co-amplification in intimal type might help *MDM2* immunohistochemistry results since 10% to 20% of these tumours are MDM2-negative. Concomitant amplification of *KIT* and *MDM2/4* may be associated with resistance to anti-cancer immune therapy; thus, the importance of adequately providing the *MDM2* assays [28,36]. The data we identified across our methods were gathered from 7/23 patients to whom *MDM2* analysis was provided in terms of immunohistochemistry. *MDM2* positive status was confirmed in two [38,40] out of the three patients, one being found MDM2-negative [30], and *MDM2* amplification in four out of the five subjects that were analysed [24,26,27,43]. Of note, one 50-year-old male had both *MDM2* assays performed, and *MDM2* amplification was negative, while *MDM2* immunostaining was positive [38] (Table 2).

To summarize, studies varying from case reports to nationwide registries of cardiac sarcoma patients represent ongoing work in finding the best strategy to classify and describe these tumours based on genomic profiles, histological traits, immune phenotypes, and molecular characterization; the core role in addressing primary intimal sarcomas being played by *MDM2* status [85,86,87,88,89,90].

### 4.4. Current Limitations and Further Expansion

Based on the previously mentioned data, several aspects should be noted. They are still a matter of debate and further studies are necessary.

Firstly, there is the exceptional issue of the paediatric population diagnosed with primary cardiac sarcomas of different types [91,92,93]. The case published by Verbeek et al. [24] in 2023 introduced a tumour that displayed a positive amplification of the *MDM2* gene plus a homozygous loss of the *CDKN2A* (cyclin-dependent kinase inhibitor 2A) gene on 9p21 in a 4-year-old child. The connection to a more aggressive profile in terms of early diagnosis should be performed; yet the subject had a remarkable 8-year free disease interval after complete tumour removal [24]. Moreover, younger adults might present with this neoplasia (according to our case-focused analysis, the youngest patients were 34 [43] and 35 years [35]). Also, the study based on the SEER database with respect to cardiac sarcomas identified 17% of them in patients younger than 30 years [28].

Secondarily, there is the question of intimal sarcoma as a second tumour in one patient or tumour progression from prior neoplasia (apart from the complex chapter of radiation-induced sarcomas) [94,95,96,97]. In this instance, we mention the case published by Domínguez-Massa et al. [33], which showed a 66-year-old woman who underwent two cardiac surgeries. The first one identified a neoplasia of uncertain origin, namely, an inflammatory myofibroblastic tumour, but three years later, an intimal sarcoma was identified. Whether this comes as a tumour progression, or whether they belong to distinct clusters, is still unknown [33].

A third point is represented by a brief mention of the time frame of papers we accessed amid the recent COVID-19 pandemic. Multiple medical and surgical entities (including in the cardio-thoracic field) required a re-assessment considering a positive infection with coronavirus to be taken into consideration as a differential diagnosis [98,99,100,101] on first admission of a subject that had a recent onset of fever and dyspnoea, as also seen in rapidly progressive intimal sarcomas [22,39,42].

A fourth insight involves the metastatic profile in cardiac intimal sarcomas. One-third of the subjects (*n* = 23) had distant spreading (brain and bone were more frequently reported), but one patient (hence, representing 16.6% of this subgroup) had unilateral adrenal metastasis [26,30,37,38,43,44]. Moreover, the SEER database on cardiac sarcomas also showed that one-third of them were associated with distant metastases, most frequently being pulmonary followed by bone. Brain metastasis was associated with the most severe outcome (*p* = 0.018) [28]. As opposed to liver, lung, cerebral, and osseous secondary involvement, adrenals [44] are less likely to be affected (mostly, in cases with a multi-organ disseminated disease), and this should be differentiated from adrenal incidentalomas underlying non-functioning cortico-adrenal adenomas based on their hormonal profile [102,103,104]. Adrenal biopsy should be performed upon pheochromocytoma exclusion, and it is used only in selected cases. PET-CT might show hypermetabolic uptake, while surgical resection is preferred in single large adrenal tumours that require histological confirmation [105,106,107,108].

## 5. Conclusions

Primary cardiac intimal sarcomas involve a heterogeneous picture from presentation to molecular findings in addition to multi-layered strategies of diagnosis and therapy. *MDM2* assays remain at the core of evaluating these tumours. This novel case adds to the current limited publications with respect to this particular histological type. The choice of performing an ultrasound-guided transthoracic core needle biopsy in a patient who was not a surgery candidate, the issue of having a post-radiation sarcoma following prior therapy for mammary cancer and the fulminant outcome, makes the case remarkable, and awareness represents the key factor. This adds to the sample-focused analysis of twenty-three cases amid latest 39 months, according to our methods of research in addition to re-do *MDM2* profiling in another 35 patients across two studies and another cohort evaluating the outcome in cardiac sarcomas. Overall, these data point out a challenging condition with a very aggressive outcome, while MDM2/*MDM2* assays might help the understanding of these tumours to a better choice of therapy.

## Figures and Tables

**Figure 1 diagnostics-14-00919-f001:**
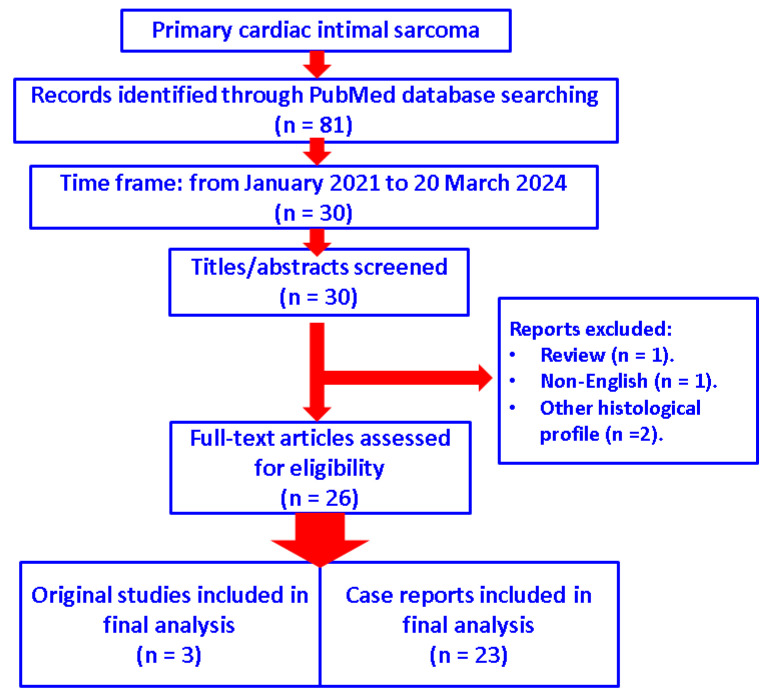
Diagram showing the flow of research according to our methods (*n* = number of studies).

**Figure 2 diagnostics-14-00919-f002:**
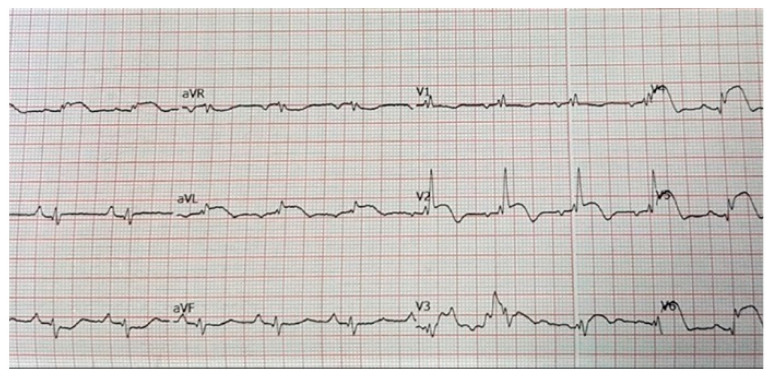
Electrocardiogram of a 65-year-old female patient without prior history of cardiovascular disease: normal sinus rhythm with a resting heart rate of 69 bpm, poor R wave progression, right bundle branch block ST elevation in leads V2–V6, I, a VL, and reciprocal ST depression in leads DIII and aVR, biphasic T waves in leads V2–V3.

**Figure 3 diagnostics-14-00919-f003:**
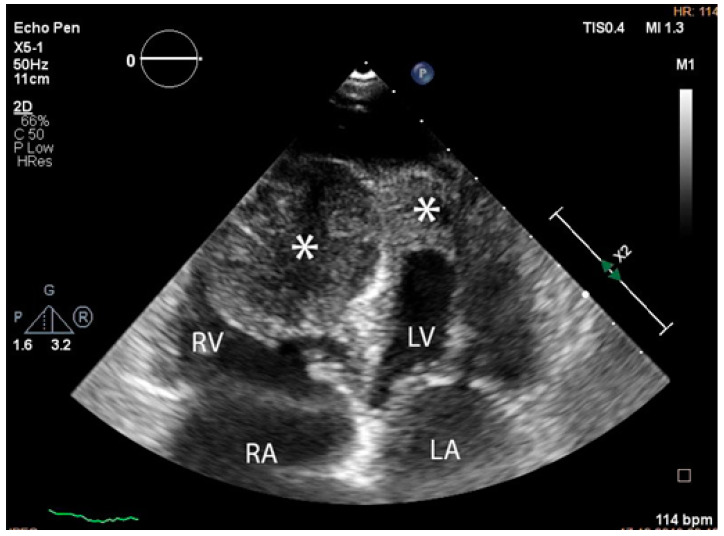
Transthoracic echocardiographic apical four-chamber view: a larger mass (∗) in the right ventricle which also involves the apex of the left ventricle. (Abbreviations: RA = right atrium, LA = left atrium, RV = right ventricle, LV = left ventricle, (∗) cardiac mass.)

**Figure 4 diagnostics-14-00919-f004:**
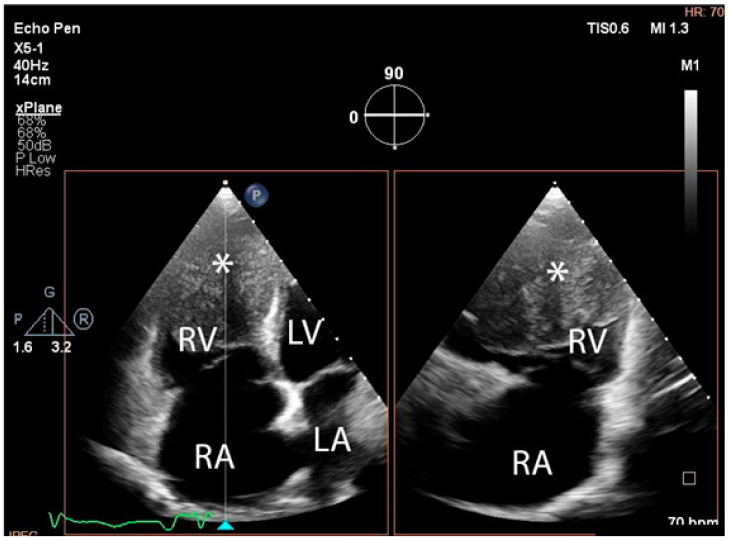
Transthoracic echocardiographic, biplane view of the right ventricle; note the mass (∗) that displaces a significant part of the cavity. (Abbreviations: RA = right atrium, LA = left atrium, RV = right ventricle, LV = left ventricle, (∗) cardiac mass.)

**Figure 5 diagnostics-14-00919-f005:**
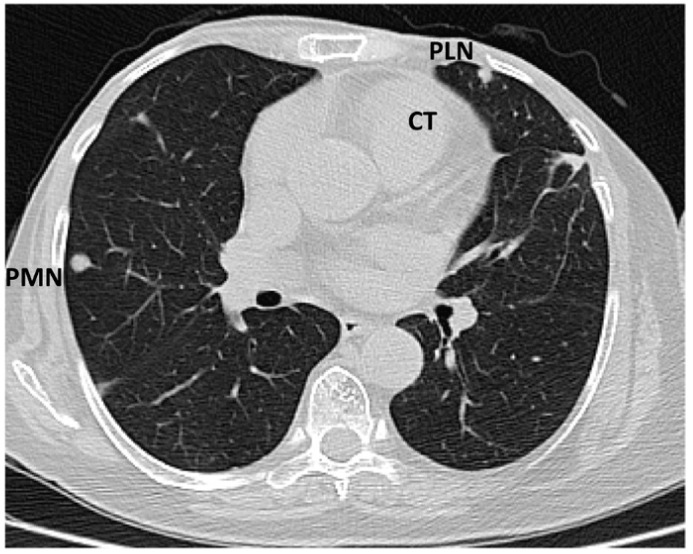
Computed tomography scan: sub-pleural peripheral nodular lesions typical of tumours (abbreviations: CT = cardiac tumour, PLN = pleural nodule, PMN = pulmonary nodule).

**Figure 6 diagnostics-14-00919-f006:**
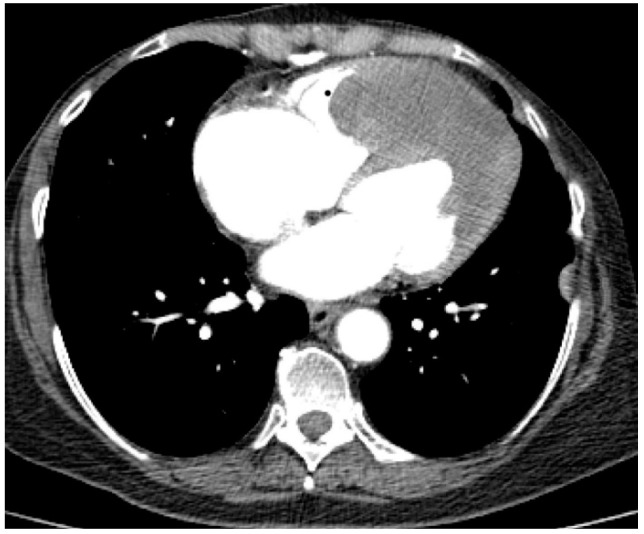
Computed tomography scan: parietal tumour thickening at the level of the apex of the heart predominantly in the left ventricle, slightly).

**Figure 7 diagnostics-14-00919-f007:**
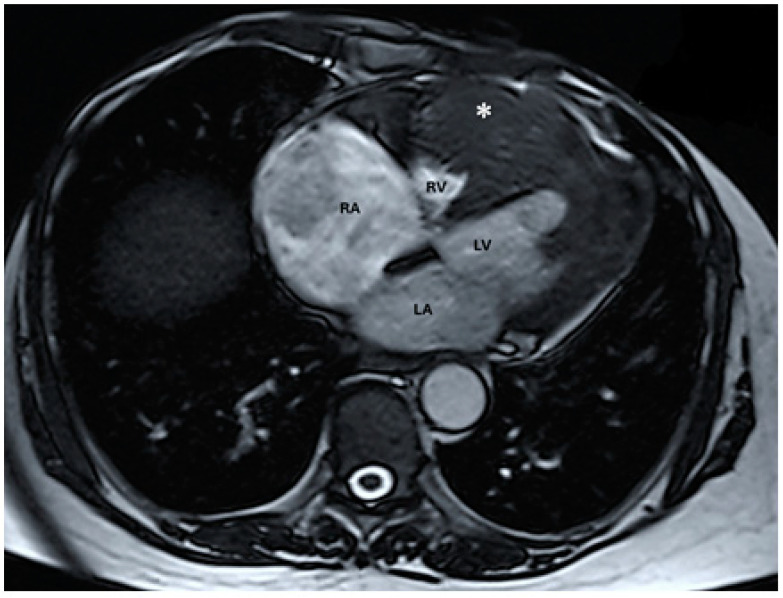
Native magnetic resonance imaging showing parietal thickening at the level of the left ventricle with discreet T2 hyper signal (abbreviations: RA = right atrium, LA = left atrium, RV = right ventricle, LV = left ventricle, (∗) cardiac mass).

**Figure 8 diagnostics-14-00919-f008:**
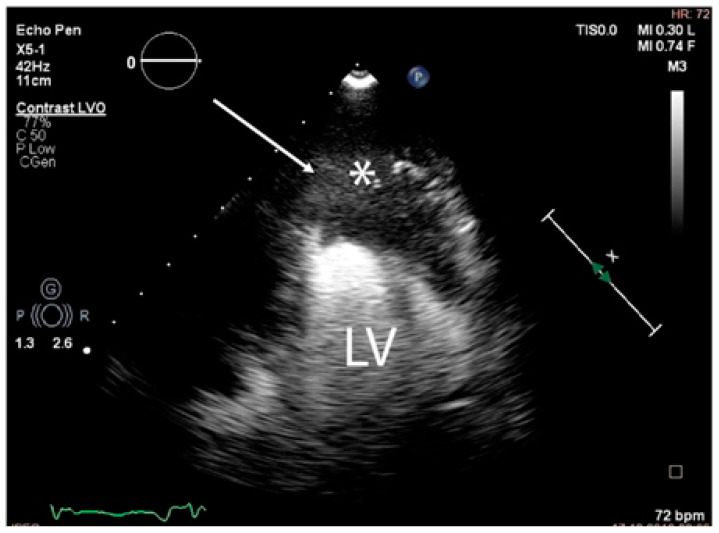
The trajectory (arrow) of the ultrasound-guided puncture at the level of the apex of the heart (abbreviations: LV = left ventricle, (∗) cardiac mass).

**Figure 9 diagnostics-14-00919-f009:**
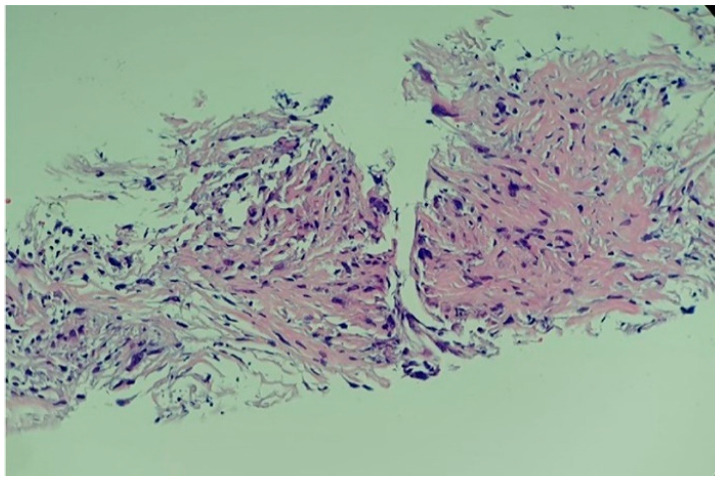
Histological report of the high-grade primary intimal cardiac sarcoma: hyper-cellular tumour with marked pleomorphic and elevated mitotic rhythm (haematoxylin–eosin; 20X).

**Figure 10 diagnostics-14-00919-f010:**
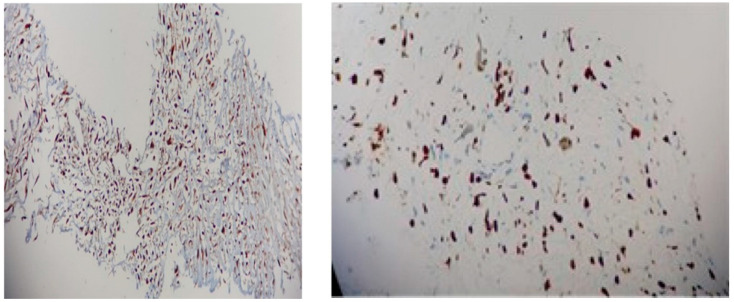
Immunohistochemistry report: (**left**) *MDM2* with multifocal nuclear positivity (20X); (**right**) Ki67 of 80% (20X).

**Table 1 diagnostics-14-00919-t001:** Case- and study-focused analysis on published data amid the latest 39 months according to our methods; the article order starts with the most recent publication date [19,20,21,22,23,24,25,26,27,28,29,30,31,32,33,34,35,36,37,38,39,40,41,42,43,44].

First Author. Publication Year. Reference.	Patient. Presentation. Diagnosis Tools.	Intervention. Outcome.
Martinho 2024 [19]	Male, 38 years old. Acute right heart failure. Diagnosis after cardiac surgery (primary intimal cardiac sarcoma; tumour of 4.2 cm).	Early post-operatory recurrence (invasion in mitral valve and pulmonary veins) → inoperable state → palliative care: pazopanib for 13 months → disease progression after 13 months → died 18 months since initial diagnosis.
Liu 2024 [20]	Male, 43 years old. Admission for facial and peripheral oedema (2 months). Transthoracic echocardiography: tumour of 9 cm (giant right ventricular sarcoma). Mild pre-operative thrombocytopenia. Confirmation (primary intimal cardiac sarcoma) after cardiac surgery.	Tumour debulking resection under cardiopulmonary bypass → immediate after surgery: severe coagulopathy and inflammatory syndrome → stabilized and discharged after 18 days.
Utsunomiya 2023 [21]	Female in her 70s. Dyspnoea and palpitations (6 months). Admission for severe mitral regurgitation and pulmonary hypertension. Transthoracic echocardiography, CT, MRI: left atrial tumour > 9 cm. ^18^F-FDG PET-CT: cardiac tracer uptake at tumour level → suspicion of primary cardiac sarcoma.	Surgery: tumour resection + aortic and mitral valve replacements → post-operatory histological diagnosis → no recurrence during the 20 months following resection.
Bergonzoni 2023 [22]	Male, 73 years old. Clinical and ultrasound features mimicking an infective bacterial endocarditis of the mitral valve.	Surgical resection of the tumour → post-operatory confirmation of mitral valve intimal sarcoma → anthracycline-based chemotherapy → rapid liver and pulmonary spreading → died in 11 months.
Awoyemi 2023 [23]	Female, 68 years old. History of treated breast cancer + melanoma. Admission for chest pressure and dyspnoea. Transthoracic echocardiography: tumour of 3 cm (right ventricle) → MRI (tumour of 8 cm).	Right heart catheterization with endo-myocardial biopsy → suggested a liposarcoma but confirmed with cardiac intimal sarcoma → multimodal imagery staging → decision of radiotherapy (surgery was prohibited because of tumour size + the patient refused chemotherapy) → pericardial effusion with tamponade → successful palliative pericardiocentesis → palliative pericardial window.
Verbeek 2023 [24]	Child, 4 years old. Admitted for right heart failure due to cardiac intimal sarcoma in the right ventricle.	Urgent surgical resection (within 24 h since first admission) → complete tumour resection + De Vega plasty of tricuspid → post-operatory exam (tumour of 3.5 cm) → PET-CT: hypermetabolic uptake in multiple lymph nodes → chemotherapy was refused by the parents → no recurrence during the 8-year follow-up.
Cho 2023 [25]	Retrospective study on primary cardiac sarcomas (*n* = 48) amid *MDM2* testing. N1 = 15 primary cardiac intimal sarcomas (seven new cases were identified based on MDM2-positive status). Average age: 47.2 (12–78) years.	MDM2-positive was correlated with a better outcome than sarcomas with MDM2-negative status.
Ballout 2023 [26]	Male, 35 years old. Admission due to neurological symptoms caused by brain metastasis. CT: brain metastasis and left atrial tumour. Transthoracic echocardiography: a cardiac tumour of 7 cm.	Decision of cardiac tumour removal first → uncomplicated cardiac surgery → intimal sarcoma confirmation → resection of brain metastasis after 8 days → confirmation of similar histological report (sarcoma metastasis) → planned chemotherapy (ifosfamide) + 6-weeek radiotherapy → 6-month chemotherapy (ifosfamide + doxorubicin).
Mahdi 2023 [27]	Male, 81 years old. Admission for dyspnoea (several months). History of melanoma. Transthoracic echocardiography: tumour of 3.2 cm + insufficiency and stenosis of the pulmonary valve → CT angiography → MRI: tumour was attached to pulmonary artery → confirmation via transoesophageal echocardiogram.	Surgical resection (with reconstruction of main pulmonary artery + pulmonary valve) → confirmation of pulmonary artery intimal sarcoma → oncologic surveillance.
Khan 2023 [28]	Study on cardiac sarcomas (SEER database, respectively, genomic data were extracted from the TCGA database). Data from 2000 to 2018. *n* = 400 patients with (all types) cardiac sarcomas (mean age of 47.7 ± 18.3 years). Cause-specific survival rate of 13%.	Surgery was associated with a better outcome than chemotherapy. Younger age at diagnosis (<50 years) better outcome than >50 years. Intimal sarcoma (origins in pulmonary trunk, artery, and vein) was included in general analysis (mostly, were primary cardiac angiosarcomas).
Nakashima 2023 [29]	Female, 57 years old. Admission: acute myocardial infarction caused by thrombotic occlusion of right coronary artery.	Percutaneous coronary intervention of the right coronary artery → confirmation of coronary artery intimal sarcoma → surgical resection + coronary artery bypass surgery of the artery → adjuvant chemotherapy for 1 year → focal recurrence after 3 years (caudal area of left ventricular inferior wall) → radiotherapy → after 4 years, negative uptake at PET-CT → no recurrence after 7 years since surgery (first report of coronary artery intimal sarcoma with long-term survival following multimodal therapy).
Ye 2023 [30]	Female, 52 years old. Admission for 3-month history of cough and dyspnoea. History of cervical cancer. Transthoracic echocardiography: tumour of 6.7 cm in left atrium → CT, MRI to exclude metastasis.	Surgical resection → high-grade intimal sarcoma confirmation with MDM2-negative status (Ki67 of 40%) at immunohistochemistry → after 4 months → bone metastasis → chemotherapy.
Lloyd 2023 [31]	Male, 57 years old. Admission for ischemic stroke. CT: left atrial tumour (on second admission).	Surgical resection → confirmation of primary intimal sarcoma → radiotherapy (the patient refused chemotherapy) → died after 14 months.
Yafasova 2023 [32]	Male, 60 years old. History of prior aortoplasty for congenital aortic stenosis + mechanical aortic valve replacement and aortic stenting for aortic dilatation. Admission for dyspnoea, night sweats, malaise. Transthoracic and transoesophageal echocardiography + CT: left atrial tumour (thrombus suspicion) → PET-CT: hypermetabolic status.	Non-radical mass resection + insertion of biological valve prostheses → confirmation of intimal sarcoma → died after 3 months.
Domínguez-Massa 2023 [33]	Female, 66 years old. Intimal sarcoma was the second cardiac tumour (tumour progression?).	Surgical resection of left atrial tumour → confirmed as inflammatory myofibroblastic tumour → after 3 years: second surgery for a tumour in right atrium → intimal sarcoma.
Salazar 2022 [34]	Female, 65 years old. Admission for 3-month history of dyspnoea, asthenia, anorexia, and weight loss. Transthoracic echocardiogram: tumour in auricles + trans-valvular obstruction of mitral valve. CT: large mass in inter-auricular septum.	Trans-venous biopsy: confirmation of primary intimal sarcoma → the patient was not a candidate for surgery → palliative chemotherapy → died in 2 weeks.
Chang 2022 [35]	Male, 48 years old. Admission for 3-month history of palpitation and chest pressure. Echocardiography: tumour of 5.8 cm → PET-CT: hypermetabolic multiple areas → after 2 months: clinical aggravation.	Percutaneous intravenous catheter biopsy: confirmation of primary intimal sarcoma → exploratory immunotherapy: immune checkpoint inhibitor PD-1 antibody for 5 cycles → improvement → continued for 13 cycles.
Yamada 2022 [36]	Retrospective analysis of 20 cases with intimal sarcoma. Histological report: necrosis (75%); fibrinous deposits (70%); myxoid stroma (60%); haemorrhage (60%); abnormal mitosis (50%); and cartilaginous pattern (5%).	Immunohistochemistry: focal positive for *MDM2* (80%). FISH: *MDM2* amplification (55%), *PDGFRA* amplification (5%). *MDM2* amplification correlated with myxoid pattern (*p* = 0.0194).
Ho 2021 [37]	Female, 37 years old. Admission for paroxysmal atrial fibrillation and left atrial mass. Transoesophageal echocardiogram: left atrial mass (suspicion for thrombus) → MRI: cardiac mass of 4.7 cm.	Complete surgical resection of left atrial tumour→ 1 year later: back pain due to T10 and L1 pathological fractures → vertebral body biopsy → vertebroplasty → diagnosis of metastatic leiomyosarcoma → MRI: spinal canal encroachment → re-do analysis: metastatic intimal cardiac sarcoma → 6 weeks of radiotherapy → dyspnoea due to paraspinal metastases → palliative chemotherapy (adriamycin/Ifosfamide).
Chiarelli 2021 [38]	Male, 50 years old. Admission for bowel obstruction due to metastatic intimal sarcoma.	Incomplete surgical resection after 1 + 1/2 years since admission of left atrium tumour → chemotherapy (doxorubicin + isophosphamide) → post-operatory local recurrence after 4 months → radiotherapy → 18 months later: ileum metastasis → radiotherapy (no tolerance to chemotherapy).
Diamond 2021 [39]	Female, 51 years old. Admission for functional mitral stenosis (constitutional symptoms, obstructive shock); 1-day history of dyspnoea, sudden fatigue, headache, and fever. Transthoracic echocardiogram: left atrial mass with intermittent obstruction of left ventricular inflow → hypotension precipitated acute tubular necrosis → renal failure → transoesophageal echocardiogram: left atrial mass of 4 cm → brain MRI: multiple emboli → recurrent paroxysmal atrial fibrillation → amiodarone.	Surgical resection of the left atrial mass: confirmation of intimal sarcoma → proton beam therapy.
Nakagawa-Kamiya 2021 [40]	Male, 45 years old. Ten-month history of cough, palpitation, weight loss, hyperhidrosis. Transthoracic echocardiography: left atrium tumour of 3.7 cm → confirmed at transthoracic echocardiography.	Emergency surgical resection: confirmation of intimal sarcoma upon MDM2-positive immunostaining.
Pyo 2021 [41]	Female, 55 years old. (First) admission for dyspnoea and shock. Transthoracic echocardiography and CT: left atrial tumour of 3.9 cm (pulmonary vein invasion).	First surgery: complete tumour resection (auto-transplantation technique) + replacement of mitral valve (mechanical prosthetic) → confirmation of intimal sarcoma → local recurrence after 6 months (CT). Second surgery: novel surgical approach: resection of left atrial wall and pulmonary veins → neo-left atrium by using lung hilum and posterior mediastinal soft tissue (no further reconstructive procedures) → adjuvant radiotherapy was declined by the patient → recurrence after 6 months → died after 8 months since second surgery.
Todo 2021 [42]	Male, 78 years old. Admission for heart failure caused by severe mitral regurgitation. Transoesophageal echocardiography: suspicion of non-bacterial thrombotic endocarditis (+ left atrial mass).	Surgery: semi-urgent mitral valve replacement + tumourectomy (due to refractory hear failure) → confirmation of primary intimal cardiac sarcoma.
Püsküllüoglu 2021 [43]	Female, 34 years old. Admission for heart failure (NYHA class II) + weakness. Intraoperative transoesophageal echocardiography: left atrium mass.	Open heart surgery→ post-operatory confirmation of intimal sarcoma → symptomatic brain metastasis after 2 months → palliative radiotherapy + chemotherapy (doxorubicin) → the patient soon died.
Durieux 2021 [44]	Male, 37 years old male. Admission for dynamic obstruction of the mitral valve (NYHA classes II-III, anorexia, epigastric pain—for several days). Transoesophageal echocardiography: left atrium mass (of 4 cm) → MRI confirmation.	Surgery: minimally invasive thoracoscopic approach (incision in right fourth intercostal space) → initial misdiagnosed as myxoma →local recurrence after 3 months → second surgery → intimal sarcoma → adjuvant chemotherapy → after 1 year PET-CT showed left adrenal metastasis → surgical removal.

Abbreviations: CDKN2A = cyclin-dependent kinase inhibitor 2A; CT = computed tomography; ^18^F-FDG PET-CT = ^18^F-fluorodeoxyglucose positronic emission tomography–CT; L1 = lumbar vertebra; *MDM2* = mouse double minute 2 homolog; MRI = magnetic resonance imaging; NYHA = New York Heart Association; PD-1 = Programmed Cell Death Protein 1; PDGFRA = platelet-derived growth factor receptor alpha; SEER = The Surveillance, Epidemiology, and End Results; T10 = thoracic vertebra.

**Table 2 diagnostics-14-00919-t002:** MDM2 analysis according to our sample-based research [24,26,27,38,43].

First Author. Year of Publication. Reference Number.	Patient.	MDM2 Status (*MDM2* Genetic Analysis and/or *MDM2* Immunohistochemistry).
Verbeek 2023 [24]	Child, 4 years old.	*MDM2* amplification + homozygous loss of *CDKN2A* on 9p21 in tumour cells.
Ballout 2023 [26]	Male, 35 years old.	FISH: *MDM2* amplifications.
Mahdi 2023 [27]	Male, 81 years old.	*MDM2* amplification.
Ye 2023 [30]	Female, 52 years old.	MDM2-negative status (Ki67 of 40%) at immunohistochemistry.
Chiarelli 2021 [38]	Male, 50 years old.	MDM2-positive. FISH: *MDM2* amplification negative.
Nakagawa-Kamiya 2021 [40]	Male, 45 years old.	MDM2-positive immunostaining.
Püsküllüoglu 2021 [43]	Female, 34 years old.	*MDM2* amplification in tumour cells (in situ hybridization).

Abbreviations: *MDM2* = mouse double minute 2 homolog; FISH = fluorescent in situ hybridization.

## Data Availability

Not applicable.

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
