# Peer review of "Primary Cardiac Intimal Sarcoma: Multi-Layered Strategy and Core Role of MDM2 Amplification/Co-Amplification and MDM2 Immunostaining"

_diagnostics, 2024, doi:10.3390/diagnostics14090919_

Round 1
Reviewer 1 Report
Comments and Suggestions for Authors
add pathologist. H&E photomicrograph of low power and poor quality. WES or WGS not mentioned for these tumors. mention WHO criteria.
Comments on the Quality of English LanguageWhat does this mean on line 85-6? "a fine index of suspicion and also including a multimodal imagery panel in association with tools of direct tumor access which are mandatory to a better outcome"
Author Response
Response to Review 1 Comments
Dear Reviewer,
Thank you very much for your time and your effort to review our manuscript.
We are very grateful for providing your valuable feedback on the article.
Here is our response and related amendment that has been made in the manuscript according to your review (marked in yellow color).
Add pathologist. H&E photomicrograph of low power and poor quality. WES or WGS not mentioned for these tumors. Mention WHO criteria.
Thank you very much. The general frame according to WHO has been introduced: “According to the World Health Organization (WHO), angiosarcomas are the most common type of sarcoma (40%) followed by rhabdomyosarcoma (5%) and, less frequently, liposarcoma, osteosarcoma, leiomyosarcoma, as well as fibrosarcoma. Intimal sarcomas of primary cardiac origin are extremely rare mesenchymal tumours across these mentioned histological types”
The WES or WGS is not available for the case in point. All the relevant data with this particular regard (if any) in the matter of literature review has been introduced in Table 1 and 2.
We respectfully mention that the H&E captures are the best we could achieve. The editorial team will decide if they remain as supplementary file or within main manuscript. Thank you
The pathologist amid the multidisciplinary team of evaluation has been mentioned: “The indications for the improvement of the treatment strategies are correctly defined by the oncologic boards that include the oncologist, the surgeon, the cardiologist, the radiologist, the radiotherapist, the pathologist, “
Moreover, the cases mentioned in Table 1 across out research were based on the specific confirmation of primary intimal cardiac sarcoma.
With regard to the case on point the data were provided as following: “Histological reports confirmed the presence of a high grade primary intimal cardiac sarcoma with areas of myxoid differentiation and epithelioid-appearing cells.”’ “hyper-cellular tumour with marked pleomorphic and elevated mitotic rhythm”;“Immunohistochemistry analysis revealed a positive reaction for MDM2 and a Ki67 proliferation marker of 20% to 80% in tumour cells. The samples were found negative with regard to myogenin, CD31, CD34, D2-40, and desmin.”
Thank you very much
Comments on the Quality of English Language. What does this mean on line 85-6? "a fine index of suspicion and also including a multimodal imagery panel in association with tools of direct tumor access which are mandatory to a better outcome"
Thank you. We corrected the statement. It means that the low index of clinical suspicion is followed by the mentioned panel of complex assessments. “The management of primary cardiac tumours is complex, starting with a fine index of suspicion and followed by a multimodal imagery panel in association with tools of direct tumour access which are mandatory to a better outcome”…
Thank you very much.

Reviewer 2 Report
Comments and Suggestions for Authors
Cardiac intimal sarcomas are a grave, relative rare entity that troubles cardiologists, cardiac surgeons and oncologists. The presentation in form of an extensive review article is important, and is well supported by the authors. The focus on diagnosis based on MDM2 amplification/co-amplification and MDM2 immunostaining is clinically very helpful. I therefore recommend the publication of the paper by the journal. Minor language editing is necessary. I stand at your disposal for further information.
Comments on the Quality of English LanguageMInor language editing by a native English language speaker is necessary.
Author Response
Response to Review 2 Comments
Dear Reviewer,
Thank you very much for your time and your effort to review our manuscript.
We are very grateful for your insightful comments and observations, also, for providing your valuable feedback on the article.
Here is a point-by-point response and related amendments that have been made in the manuscript according to your review (marked in yellow color).
Cardiac intimal sarcomas are a grave, relative rare entity that troubles cardiologists, cardiac surgeons and oncologists. The presentation in form of an extensive review article is important, and is well supported by the authors. The focus on diagnosis based on MDM2 amplification/co-amplification and MDM2 immunostaining is clinically very helpful. I therefore recommend the publication of the paper by the journal. Minor language editing is necessary. I stand at your disposal for further information.
Thank you. We revised the English language.
Thank you very much.

Reviewer 3 Report
Comments and Suggestions for Authors
Primary cardiac tumours are uncommon, with 75% being benign and the remaining 25% malignant, mainly sarcomas. This paper reviews primary cardiac intimal sarcoma (PCIS), a rare malignancy, focusing on recent data and multidisciplinary management. It emphasizes the importance of MDM2 immunostaining and genetic analysis for diagnosis and intervention. A study reviewing cases from 2021 to 2023 identified 23 patients, presenting with non-specific symptoms like heart failure and dyspnoea. Misdiagnoses were common, and some patients had non-cardiac malignancies or distant metastasis. Imaging tools like CT and PET-CT are crucial for diagnosis. The paper highlights the role of MDM2 amplification in diagnosis and suggests further research on its prognostic value in PCIS management.
General comments
This is a manuscript addressing “Primary Cardiac Intimal Sarcoma: Multi-Layered Strategy and Core Role of MDM2 Amplification/Co-Amplification and MDM2 Immunostaining”. This is a well-written paper. However, the reviewer has some concerns need to be addressed.
Specific comments
1) Line 247. The sentence has some inconsistencies and unclear references: The term "respectively" is used ambiguously. It should clearly correlate specific elements to avoid confusion. In this case, it seems to attempt to link "neoadjuvant" to "N = 4/48" and "adjuvant" to "N = 34/48", but the placement and punctuation make this unclear. A clearer way to phrase this might be: "Complete tumour resection (N = 11/48), partial excision (N = 23/48), cardiac biopsy or pericardiotomy (N = 10/48), cardiac transplant (N = 4/48), neoadjuvant chemotherapy or radiotherapy (N = 4/48), and adjuvant chemotherapy or radiotherapy (N = 34/48)." Sentences containing "respectively" should be revised for clarity. Ensure that "respectively" is used correctly to match items in two parallel lists in the same order.
2) In the section titled "A Novel Case Identification" within the Discussion, the report of a case should be moved to the Results section because it involves the presentation of a specific case.
3) In Line 319, mention that the larger mass is not shown in Figure 3. In Line 322, clarify that the mass (indicated by *) is not shown in Figure 4.
4) In Figure 7, ensure that the labels for the left atrium (LA) and right ventricle (RV) are correctly indicated.
5) Provide an explanation for the symbol (*) wherever it is used in the text or figures.
Author Response
Response to Review 3 Comments
Dear Reviewer,
Thank you very much for your time and your effort to review our manuscript.
We are very grateful for your insightful comments and observations, also, for providing your valuable feedback on the article.
Here is a point-by-point response and related amendments that have been made in the manuscript according to your review (marked in yellow color).
Primary cardiac tumors are uncommon, with 75% being benign and the remaining 25% malignant, mainly sarcomas. This paper reviews primary cardiac intimal sarcoma (PCIS), a rare malignancy, focusing on recent data and multidisciplinary management. It emphasizes the importance of MDM2 immunostaining and genetic analysis for diagnosis and intervention. A study reviewing cases from 2021 to 2023 identified 23 patients, presenting with non-specific symptoms like heart failure and dyspnea. Misdiagnoses were common, and some patients had non-cardiac malignancies or distant metastasis. Imaging tools like CT and PET-CT are crucial for diagnosis. The paper highlights the role of MDM2 amplification in diagnosis and suggests further research on its prognostic value in PCIS management.
Thank you very much.
General comments. This is a manuscript addressing “Primary Cardiac Intimal Sarcoma: Multi-Layered Strategy and Core Role of MDM2 Amplification/Co-Amplification and MDM2 Immunostaining”. This is a well-written paper. However, the reviewer has some concerns need to be addressed.
Thank you very much. We addressed the mentioned concerns as follows.
Specific comments
Line 247. The sentence has some inconsistencies and unclear references: The term "respectively" is used ambiguously. It should clearly correlate specific elements to avoid confusion. In this case, it seems to attempt to link "neoadjuvant" to "N = 4/48" and "adjuvant" to "N = 34/48", but the placement and punctuation make this unclear. A clearer way to phrase this might be: "Complete tumour resection (N = 11/48), partial excision (N = 23/48), cardiac biopsy or pericardiotomy (N = 10/48), cardiac transplant (N = 4/48), neoadjuvant chemotherapy or radiotherapy (N = 4/48), and adjuvant chemotherapy or radiotherapy (N = 34/48)." Sentences containing "respectively" should be revised for clarity. Ensure that "respectively" is used correctly to match items in two parallel lists in the same order.
Thank you very much. We corrected them according to your recommendations: “Across the study of Cho et al. [25] (N = 48) multimodal management included: complete tumor resection (N = 11/48), partial excision (N = 23/48), cardiac biopsy or pericardiotomy (N = 10/48), cardiac transplant (N = 4/48), neoadjuvant chemotherapy or radiotherapy (N = 4/48), and adjuvant chemotherapy or radiotherapy (N = 34/48); neoadjuvant (N = 4/48), and adjuvant chemotherapy or radiotherapy (N = 34/48).” Thank you very much.
In the section titled "A Novel Case Identification" within the Discussion, the report of a case should be moved to the Results section because it involves the presentation of a specific case.
Thank you. We respectfully mention that the article is firstly a complex review and the data were focused around the sample-based analysis with a recent time frame of publication. The case on point has been introduced only as a base of discussion due to the rarity of such situation, but the key findings are the data from prior published papers. Thank you very much.
In Line 319, mention that the larger mass is not shown in Figure 3.
Thank you very much. We really appreciate it. This is a very important aspect and we corrected it as following:
“Figure 3. Transthoracic echocardiographic apical four-chamber view: a larger mass (*) in the right ventricle which also involves the apex of the left ventricle. (Abbreviations: RA=right atrium, LA=left atrium, RV=right ventricle, LV=left ventricle, (*) cardiac mass)”
Thank you very much
In Line 322, clarify that the mass (indicated by *) is not shown in Figure 4.
Thank you. This is very important and we are grateful for this observation. We corrected it as following:
“Figure 4. Transthoracic echocardiographic, biplane view of the right ventricle; note the mass (*) which displaces a significant part of the cavity. (Abbreviations: RA=right atrium, LA=left atrium, RV=right ventricle, LV=left ventricle, (*) cardiac mass).”
Thank you very much
In Figure 7, ensure that the labels for the left atrium (LA) and right ventricle (RV) are correctly indicated.
Thank you. We really appreciate it! We corrected the labels as followings. Thank you very much
“Figure 7. Native magnetic resonance imaging showing parietal thickening at the level of the left ventricle with discreet T2 hyper signal (Abbreviations: RA=right atrium, LA=left atrium, RV=right ventricle, LV=left ventricle, (*) cardiac mass).”
Provide an explanation for the symbol (*) wherever it is used in the text or figures.
Thank you very much. We did the corrections as we pointed above and we appreciate all your recommendations.
Thank you very much

Round 2
Reviewer 3 Report
Comments and Suggestions for Authors
The authors have corrected the manuscript according to the reviewer's comments.